# Organoboron Complexes as Thermally Activated Delayed Fluorescence (TADF) Materials for Organic Light-Emitting Diodes (OLEDs): A Computational Study

**DOI:** 10.3390/molecules28196952

**Published:** 2023-10-06

**Authors:** Jamilah A. Asiri, Walid M. I. Hasan, Abdesslem Jedidi, Shaaban A. Elroby, Saadullah G. Aziz, Osman I. Osman

**Affiliations:** 1Chemistry Department, Faculty of Science, King Abdulaziz University, Jeddah 21589, Saudi Arabia; jmohammedasiri@stu.kau.edu.sa (J.A.A.); whassan@kau.edu.sa (W.M.I.H.); ajedidi@kau.edu.sa (A.J.); saziz@kau.edu.sa (S.G.A.); 2Department of Chemistry, College of Arts and Sciences, Prince Sattam bin Abdulaziz University, Wadi Ad-Dwasir 18510, Saudi Arabia; 3Chemistry Department, Faculty of Science, Beni-Suif University, Beni-Suif 62521, Egypt; 4Chemistry Department, Faculty of Science, University of Khartoum, Khartoum P.O. Box 321, Sudan

**Keywords:** organoboron, TADF, RISC, NTO, radiative decay rate

## Abstract

We report on organoboron complexes characterized by very small energy gaps (ΔE_ST_) between their singlet and triplet states, which allow for highly efficient harvesting of triplet excitons into singlet states for working as thermally activated delayed fluorescence (TADF) devices. Energy gaps ranging between 0.01 and 0.06 eV with dihedral angles of ca. 90° were registered. The spin–orbit couplings between the lowest excited S_1_ and T_1_ states yielded reversed intersystem crossing rate constants (K_RISC_) of an average of 10^5^ s^−1^. This setup accomplished radiative decay rates of ca. 10^6^ s^−1^, indicating highly potent electroluminescent devices, and hence, being suitable for application as organic light-emitting diodes.

## 1. Introduction

The first appearance of organic emitters was witnessed by detecting an emission of an anthracene crystal [1] in 1965, followed by the synthesis of the first organic light-emitting diode (OLED) cell composed of diamine and Alq3 as fluorescent materials [2]. Since then, OLED techniques have been used in many photonic applications [1,2,3]. The working principle states that “when an electric current is applied to organic materials with distinctive properties, light is emitted”. The emitted light may be fluorescence or phosphorescence, depending on the location of the excitons responsible for emitting light. Light emitted from the triplet excited state (T_1_) will be phosphorescence, while when it comes from the singlet excited state (S_1_), it is fluorescence [4,5]. Based on the spin–statistics rule, excitons are three times more abundant in the T_1_ state compared to those in the S_1_ state [6]. Accordingly, phosphorescent OLEDs may reach a maximum efficiency of 75%, while the fluorescent ones may yield only 25% productivity. In addition, phosphorescent OLEDs require the presence of spin–orbit coupling features. Metals such as Ir(III) and Pd(II) are usually used to achieve them. However, it has been shown that the pollution issue, along with the high cost, hamper the use of phosphorescent OLEDs [7,8].

In 2012, Adachi and coworkers [9] reported a fluorescent OLED with 100% theoretical efficiency. The excitons from the T_1_ state were transferred to the S_1_ state through a thermally activated delayed fluorescence (TADF) mechanism using 1,2,3,5-tetrakis(carbazol-9-yl)-4,6-dicyanobenzene molecule (4CzIPN). This emitter has perfectly localized HOMO and LUMO levels on the carbazole donor and the dicyanobenzene acceptor moieties, respectively. Therefore, TADF compounds were engineered to ensure that the HOMOs and the LUMOs were slightly separated energetically. This separation led to a small energy gap (∆E_ST_) between the two states within an order of 0.1 eV. It was realized that this kind of design allows for the transfer of excitons from the triplet state (T_1_) to the singlet state (S_1_) through reverse intersystem crossing (RISC) processes, followed by emission in the form of delayed fluorescence, with RISC decay rate constants of ca.10^6^ s^−1^ [9,10,11,12,13,14,15,16,17] (see Figure 1). To date, highly efficient organic light-emitting diodes (OLEDs) showing TADF mechanisms have been synthesized and/or theoretically explored [18,19,20,21,22].

The infinitesimal energy gap that leads to the easy harvesting of triplet state excitons (as a positive effect) causes very small oscillator strength (as a negative effect). Zero oscillator strength obtained in TADF emitters in theoretical results is common [23]. Balancing these factors is a cornerstone for developing TADF emitters [24].

Intramolecular charge transfer (ICT) is a key factor in improving the TADF properties. A reduction in the overlap between HOMO and LUMO orbitals is enhanced by ICT characteristics. Consequently, the emission of TADF emitters is greatly influenced by ICT strength. However, stronger charge transfer leads to a broader emission [25].

Charge transport is controlled by the dihedral angle between donors and acceptors [26]. The donor and acceptor units, which lie perpendicular to each other with a dihedral angle greater than 45°, are components of the most potent systems that achieve the ICT mechanism in TADF. That is, the donor–acceptor fragments affect the dihedral angle, which in turn dictate the singlet–triplet energy gap (∆E_ST_), and consequently, the optical properties, in particular, the oscillator strength (f) values [27,28,29,30,31,32]. It is worth mentioning that the quantum efficiency of OLEDs varies according to the types of donor–acceptor and substituted groups [33,34,35].

As we mentioned earlier, the RISC from the T_1_ to S_1_ excited state is key to the TADF mechanism. The rate constant (K_RISC_) of the RISC between these two lowest excited states can be evaluated using Marcus Theory [36]:(1)$$K=\frac{2\pi }{\hbar g}\frac{V_{\mathrm{SOC}}^{2}}{\sqrt{4{\pi \lambda k}_{B}T}}e^{\left[-\frac{{\left(\Delta E_{\mathrm{ST}}+\lambda \right)}^{2}}{4{\lambda k}_{B}T}\right]}$$

Vsoc is the spin–orbit coupling between two excited S_1_ and T_1_ states, λ is the reorganization energy associated with the low-frequency vibrations, ${\Delta E}_{\mathrm{ST}}$ is the adiabatic energy gap between the lowest S_1_ and T_1_ excited states, $k_{B}$ is the Boltzmann constant, ħ is the reduced Planck constant, g is the degeneracy factor, which is equal to 3 for K_RISC_ and 1 for K_ISC,_ and T is the absolute temperature, which is equal 300 K.

Furthermore, the optical properties of TADF emitters are mainly determined by the radiative rate (Kr) as one of the most important parameters. According to the equation below, the value of Kr depends on the energy of the S_1_ state (E_s1_) and the transition dipole moment (µ), which is related to the values of the oscillator strength [37].
(2)$$\mathrm{Kr}=\frac{({\mathrm{Es}}_{1}{)}^{3}\;ƒ(n)\;{µ}^{2}}{3\pi \;\varepsilon_{o}\;\hbar^{4}c^{3}}$$
where ε_o_, ħ, and c are the vacuum permittivity of space, the reduced Planck constant, and the speed of light, respectively. f(n) is a local-field correction factor, and the value of n is determined by the refractive index of the medium.

Hydrogen bonding is another parameter whose presence positively affects the oscillator strength. The existence of hydrogen bonds in quinoline TADF emitters has improved the full width at half height by 20%, followed by an increase in the quantum efficiency by approximately 48% [38].

It is worth noting that the nature of the excited state is essential for understanding the TADF mechanism in emitters. It often consists of a mixture of charge transfer (CT) and local excitation (LE) characteristics. In most cases, the charge transfer factor dominates the singlet excited state, while the local excitation engulfs the triplet state [39,40]. In some active species, the triplet state also patriciates in charge transfer [41].

With these views in mind, we endeavored to computationally examine donor–acceptor systems with electron-withdrawing (EW) groups that lead to the stability of the acceptor fragment depending on the strength of ICT and which eventually change the colorimetric efficiency [42]. 9,9-dimethyl-9,10-dihydroacridine (DMAC) was considered one of the most widely used donor fragments in TADF-OLEDs. DMAC connected to the electron acceptor dimesitylphenylborane fragment with a dihedral angle of 88° yielded ∆E_ST_ of less than 0.05 eV [29]. By adjusting the dihedral angle by different EW groups in the above emitter to be almost orthogonal in order to maintain a spatial separation between the frontier orbitals, thus ensuring a small energy gap, the TADF mechanism is realized.

The target molecules we intended to study were built by connecting dimesitylphenylborane as an acceptor to 9,9-dimethyl-9,10-dihydroacridine as a donor, forming 9,9-dimethyl-9,10-dihydroacridine-dimesitylphenylborane (Ac-B) as a parent species. In the acceptor moiety, we then grafted fluoride, cyanide, and nitro moieties, as electron-withdrawing groups, on positions 3 and 5 to avoid the repulsion with the boron atom [43,44], yielding Ac-B-F, Ac-B-CN, and Ac-B-NO_2_, respectively (see Figure 2).

## 2. Results and Discussion

The optimized geometries of the ground singlet (S_0_) states of the four boron complexes were obtained by using the ωB97XD functional with the 6-31G** basis set (see Appendix A). The dihedral angles of the Ac-B and Ac-B-F molecules of 79.63° and 81.28°, respectively, were in excellent agreement with the experimental orthogonal (88.4°) [29] dihedral angle between the donor and acceptor fragments of the parent Ac-B complex (see Appendix A).

Figure 3 illustrates the overlapping characteristics of the HOMOs and LUMOs based on the optimized ground state (S_0_). Spatial separations between the HOMOs and LUMOs were evident in all compounds. The HOMO is centered in the donor fragment (DMAC), while the LUMO engulfs the boron fragment as part of the acceptor unit. Slight overlaps between the HOMOs and LUMOs were shown on the phenylene bridges. This overlap was minimal in the Ac-B-CN molecule due to the strong electron-withdrawing effect of the cyanide group, which enhances the ICT property. In Ac-B-NO_2_ molecules, the HOMO includes part of the nitro group, which causes the maximum HOMO–LUMO overlap amongst the elected molecules.

Based on the optimized ground states for all emitters, the vertical energies of the three lowest singlets and triplets excited states, the vertical energy gaps, the revered intersystem crossing rate constants (K_RISC_), and the radiative decay rate constants (Kr) are listed in Table 1.

As shown in Appendix A, the orthogonal dihedral angles between the donor and acceptor fragments of the Ac-B and Ac-B-F molecules led to a small energy gap (∆E_ST_ < 0.10 eV), consistent with TADF emitters [45]. The same applied to Ac-B-CN, where the steric hindrance of the two cyanide groups at positions 3 and 5 in the phenylene moiety maintained the orthogonality between the two fragments. The decrease seen in the dihedral angle of AC-B-NO_2_ is in line with the appreciable overlap of its molecular orbitals.

The optimized geometries of the vertical and adiabatic excited singlet (S_1,_ S_2_, and S_3_) and triplet (T_1_, T_2_, and T_3_) states of the four understudy boron complexes were obtained by the Tamm–Dancoff approximation (TDA) using the ωB97XD functional with the 6-31G** basis set (see Appendix A). The energies of the S_2_ state of the four studied molecules were different from those of S_1_ by ca. 0.38 eV (see Table 1). These values indicate that the calculations of Kr were based solely on the vertical energies of the S_1_ state; i.e., the Boltzmann thermal population was excluded [32]. The registered values of Kr of the studied emitters reached the acceptable values of radiative decay rate constants of ca. 10^6^ s^−1^. These values are consistent with the experimental results of some compounds that achieved the TADF mechanism [46].

It is worth noting that the high radioactive rate constants of these supposedly TADF emitters are strengthened by reversed intersystem crossing from the excited T_1_ triplet state to the excited singlet S_1_ state [18,19,20,21,22]. This conjecture required the evaluation of the values of the RISC rate constants (K_RISC_) by estimating the spin–orbit couplings between the lowest excited singlet (S_1_) and triplet (T_1_) states using ADF software version 2013.01 [47] and Marcus theory [36]. It is clear from Table 1 that the value of K_RISC_ of the parent Ac-B complex of 3.12 × 10^4^ s^−1^ supports its experimentally confirmed TADF behavior [29]. On the one hand, the values of the spin–orbit couplings in Appendix A and the values of the K_RISC_ constants in Table 1 show the direct proportionality relationships between them. The nearly double spin–orbit coupling value of 0.66 cm^−1^ in the Ac-B-F complex compared to that of the parent Ac-B substrate of 0.38 cm^−1^ brought about an increase of about 68-fold (2.10 × 10^6^ s^−1^ versus 3.10 × 10^4^ s^−1^), while the doubling of that of Ac-B-NO_2_ of 1.41 cm^−1^ over that of AC-B-F of 0.66 cm^−1^ yielded an increase of ca. 6-fold (1.26 × 10^7^ s^−1^ versus 2.10 × 10^6^ s^−1^). Secondly, the K_RISC_ values of the substituted candidates were higher than those of the parent Ac-B complex. This fact indicates that the substitution of these EW groups on the phenylene spacer moiety enhanced the TADF behavior. On the other hand, there were inverse proportionality relationships between the values of the K_RISC_ and the reorganization energies (see Table 1 and Appendix A). For more information on the reorganization energy, refer to the Appendix A. Finally, it is worth noting that the values of the K_RISC_ rate constants parallel those of the radioactive decay constants (Kr) for all candidates except Ac-B-CN of 1.0 × 10^5^ s^−1^. This inconsistency was probably brought about by the comparatively small spin–orbit coupling value of 0.14 cm^−1^ for this contender.

The compounds Ac-B-F and Ac-B-CN showed a gradual decrease in the values of vertical S_1_, S_2_, and S_3_ state energies in comparison to those of the Ac-B compound, followed by a decrease in the energy gaps, as expected. The lower energy gap is related to the energy of the S_1_ state. That is, the decrease in the energy gap was directly proportional to the stability of the S_1_ state [42]. The strong interaction between the HOMO and LUMO orbitals in Ac-B-NO_2_ was also responsible for the slight increase in the energy gap value.

Absorption UV-Vis. Data for Ac-B, Ac-B-F, Ac-B-CN, and Ac-B-NO_2_ were simulated and are illustrated in Table 2 and Appendix A. A blue wavelength appeared (307–531 nm), and the results indicate more than one peak originating from the S_1_, S_2_, and S_3_ state transitions. A broad peak from the S_1_ states was noted at lower energy, in excellent agreement with the experimental results of the parent Ac-B [29]. Mostly, a broad peak is the result of charge transfer transitions [48]. The ICT characteristics in molecules with EW properties led to a clear red shift compared to the parent Ac-B molecule. These were related to the relatively increased charge transfer. In addition, the n-π* transition enhanced the redshift. That is, the introduction of the nitro group caused the signal to appear at 531 nm [49]. The diminished oscillator strength was clear due to the small energy gap. However, it was found that the insertion of two fluorine atoms in positions 3 and 5 on the phenylene unit led to the smallest full width at half-maximum (FWHM) [50].

The increase in the oscillator strength value by approximately 100% in Ac-B-F compared to that of Ac-B, together with the increase in the radiative decay rate, is attributed to the formation of hydrogen bonding. The results of the topological analysis based on the quantum theory of atoms in molecules (QTAIM) [50] indicated that there is a bond critical point (BCP) for the critical point (CP) (3,−1) representing the formation of hydrogen bonds between the X(X = F, N and O) atoms and the neighboring hydrogen atoms in the methyl groups. The following equation can be used to calculate the hydrogen bond strength, according to Espinosa and coworkers [51]:E_HB_ = ½ (Vr_BCP_)
where Vr_BCP_ is the electron potential at the BCP, and E_HB_ refers to the binding energy of the hydrogen bond. Hydrogen bonds with F, N, and O atoms of the Ac-B-F, Ac-B-Cn, and Ac-B-NO_2_ molecules, respectively, yielded binding energies of −2.77, −3.51, and −7.81 kJ/mol, respectively. It is clear from Figure 4 that two H··O bonds were formed in the case of the Ac-B-NO_2_ molecule, yielding a double binding energy. The overlap between the HOMO and LUMO orbitals, combined with the energy gap and hydrogen bonding, appear to play vital roles in amplifying the oscillator strength in Ac-B-NO_2_ (see Figure 4).

The energies of the HOMOs and LUMOs based on the optimized ground state (S_0_) are plotted in Figure 5. These results indicate a clear decrease in the energy of the LUMOs when the electron-withdrawing groups were introduced. The energies of the LUMOs in the Ac-B-F, Ac-B-CN, and Ac-B-NO_2_ molecules were stabilized by 0.16, 0.64, and 0.87 eV, respectively, compared to that of the Ac-B molecule. The stabilization of the LUMOs is proportional to the values of the meta-Hammett constants for the electron-withdrawing groups -F, -CN, and -NO_2_ (0.34, 0.65, and 0.71, respectively) [52].

The stabilization of the LUMOs enhanced the quality of the color characteristics, as shown in the absorption spectra of Ac-B-F, Ac-B-CN, and Ac-B-NO_2_. Despite the stability of the acceptor in Ac-B-CN, the oscillator strength was equal to zero, and the vanished oscillator strength was imputed to the declining overlap of the orbitals [25,53]. The slightly comparable energies of the HOMOs are explained by keeping the donor fragment untouched in our study.

The natural transition orbital (NTO) characteristics of all studied molecules, based on the optimized S_0_ state, were examined to probe the nature of the S_1_ and T_1_ excited states. As shown in Figure 6, the hole and electron wavefunctions of the S_1_ states were mainly distributed on the acridine donors and the borane acceptors, respectively, with some overlap on the phenylene bridge. For the T_1_ state, the hole and electron wavefunctions of the parent Ac-B spread over the acceptor and donor, respectively, while a reverse situation occurred when the EW groups (-F, -CN, and -NO_2_) were grafted on positions 3 and 5 of the phenylene moiety. These kinds of confinements indicate the presence of a mixture of charge transfer (CT) and local excitation (LE) transitions in both the S_1_ and T_1_ states [45]. For all compounds, the hole and electron wavefunctions overlapped on the phenylene bridge. Finally, the Ac-B-F, Ac-B-CN, and Ac-B-NO_2_ molecules were dominated by CT nature, and as a result, their S_1_ and T_1_ states were located close together [41] (see Figure 6, Table 1).

We also performed the adiabatic transitions of the S_1_ excited state, based on its optimized geometry. We note, in Table 3, that the energies of the S_1_ state were higher than those of the T_1_ state for the Ac-B, Ac-B-F, and Ac-B-CN species; however, it turned out that those of the S_1_ state were lower than those of the T_1_ state for Ac-B-NO_2_. All target molecules yielded adiabatic energy gaps (ΔE_ST_) ranging between 0.01 and 0.06 eV. It is worth mentioning that the adiabatic analysis yielded extremely lower radiative decay rates for all species compared to those of the vertical ones. This observation is supported by the absence of H-boning in Ac-B-F and the very small binding H-bonding energies of the Ac-B-CN and Ac-B-NO_2_ molecules.

In a further study of the excited S_1_ state based on the geometry of the S_1_ state, we also analyzed its NTOs. It is quite clear, as Figure 7 shows, that the hole wavefunctions of all studied compounds were mainly enclosed within the donor acridine moieties, while the electron wavefunctions were confined to the central phenylene–borane fragments [45]. Thus, the S_1_ states had a hybrid character merging local and charge transfer excitations, with an increase in the electron-withdrawing effect [54].

Moreover, Appendix A depict the NTO wavefunctions of the second (T_2_) and third (T_3_) excited triplet states, respectively. It is clear that for the second excited triplet (T_2_) states, the hole and electron wavefunctions of all studied compounds were overlapping, especially those of the parent Ac-B and Ac-B-F. The hole wavefunctions of Ac-B-CN and Ac-B-NO_2_ were mainly enclosed within the donor acridine moieties, while the electron ones were confined to the central phenylene spacer moieties. For the NTOs of T_3_, the hole wavefunction of the parent Ac-B spread over both the donor and acceptor moieties, while the electron one engulfed both the acceptor and spacer phenylene moiety. Comparatively, the hole wavefunctions of all substituted EW contenders spread over the acceptor group, while the electron ones were distributed over the acceptor and spacer units.

We also performed calculations to monitor, quantitatively, the nature of excited singlet (S_1_) and triplet (T_1_) states in the form of charge transfer CT) and local excitation (LE) characteristics within the donors and/or acceptors [39,40,41]. Table 4 presents the interfragment charge transfer (IFCT) percentages of the CT and LE characteristics of the lowest excited states. This hole–electron quantitative study of the amount of charge transfer between fragments was obtained using Multiwfn software [55].

It is clear from Table 4 that the S_1_ states of all substrates were characterized by high percentages of CT [55] as a consequence of the spatial separation of their HOMOs and LUMOs [41]. Moreover, the CT characters increased with the incorporation of the EW groups on the phenylene spacers compared to those of the AC-B parent complex. However, these evident spatial separations between the HOMOs and LUMOs do not necessarily lead to prevalent CT characteristics in the T states [41]. This is evidenced by the low CT character in the parent Ac-B complex T_1_ state of 22.92%, which enhanced its SOC value and a small S-T energy gap [29]. In addition, for both the lowest excited states S_1_ and T_1_, the CT characters dominated over those of the LE ones when the EW groups were inserted, leading to an evident decrease in the S-T energy gaps ($\Delta E_{ST}$) (see Table 1). Furthermore, it is noteworthy that a significant LE component (77.08%) characterized the T_1_ state of the parent Ac-B complex, whereas small LE percentages of 7.37%, 6.70%, and 8.66% distinguished the T_1_ states in the Ac-B-F, Ac-B-CN, and Ac-B-NO_2_ complexes, respectively. This means that the lowest excited states were characterized by mixtures of both CT and LE components, as criteria for showing TADF mechanism in emitters [39,40].

Furthermore, we analyzed the charge transfer (CT) and local excitation (LE) component characteristics of the second (T_2_) and third (T_3_) excited triplet states (see Appendix A). The T_2_ states of all the understudy complexes, except the fluoride contender, were characterized by more than 90% CT components. This is probably due to the weak electron-withdrawing capacity of the F atom compared to the strong EW effects of the CN and NO_2_ groups. Comparatively, the T_3_ states of the Ac-B and Ac-B-F candidates were characterized by comparable percentages of the CT and LE components, with small preferences toward the latter, while the T_3_ characteristics of the Ac-B-CN and Ac-B-NO_2_ complexes were dominated by ca. 75% LE characteristics. Again, these results are in line with the collaborative mixing of the CT and LE components necessary for confirming TADF emitters, which are suitable for application as organic light-emitting diodes.

## 3. Theoretical Method

Ab initio calculations of the aforementioned compounds were carried out using Gaussian09 software [56]. The ground states (S_0_) of Ac-B, Ac-B-F, Ac-B-CN, and Ac-B-NO_2_ molecules were optimized by density functional theory using the long-range corrected ωB97XD functional [57] and the 6-31G** [58] basis set. The geometries of the excited states and the singlet–triplet energy gaps (∆E_ST_) of the target compounds were investigated using the Tamm–Dancoff approximation (TDA) [59] for a better estimation of the singlet–triplet energy gaps using the same elected functional and basis set. The vertical (V) and adiabatic (ad) energies of all studied molecules were estimated using the optimized ground state (S_0_) for the former and the optimized excited states (S_1_ and T_1_) for the latter. Natural transition orbital (NTO) analysis was used to characterize the nature of the excited electronic states [60]. Theoretical procedures were carried out at an optimal ω value. Atom-in-molecule (AIM) analysis was used to determine the hydrogen bond in the molecules in the Multiwfn and VMD programs [61,62,63]. To estimate the spin–orbit coupling between the two first excited states, ADF calculations were performed using the Amsterdam Modeling Suite (AMS 2023) [47].

## 4. Conclusions

In conclusion, the Ac-B-F, Ac-B-CN, and Ac-B-NO_2_ substrates designed in this study by combining derivatives of acridine and borane as the donor and acceptor, respectively, and substituting the hydrogen atoms of the phenylene moiety at positions 3 and 5 by -F, -CN, and -NO_2_, respectively, as electron-withdrawing substituents, are promising TADF emitters. The orthogonal dihedral angle between the donor and acceptor of the parent Ac-B was nearly preserved in the substituted compounds. The electron-withdrawing effects reduced the energies of the S_1_ states, allowing for small adiabatic and vertical energy gaps ranging between 0.02 and 0.05 eV and giving high radiative decay rate constants of the order of 10^6^ s^−1^. The high radiative rate constants were complemented by fast reversed intersystem crossing rate constants of an average of 10^5^ s^−1^, which resulted from the spin–orbit coupling between the lowest excited S_1_ and T_1_ states. The enhanced oscillator strengths of the fluorescence of the substituted species were related to the hydrogen bond formation. The reduced radiative decay rate exhibited by Ac-B-CN is attributed to the suppression of emission from the S_1_ state by the reduction in the overlap between the HOMO and LUMO levels.

Finally, our study on the excited states of these organoboron complexes involved, mainly, substantial spin–orbit couplings between the lowest excited S_1_ and T_1_ states as radiationless processes. These ultrafast processes that originate from excited initial conditions encompass vestiges, where taking account of the explicit nuclear kinetic energy is definitely essential. This situation imposes a challenge by applying excited-state dynamics [64,65] for a future complete analysis.

## Figures and Tables

**Figure 1 molecules-28-06952-f001:**
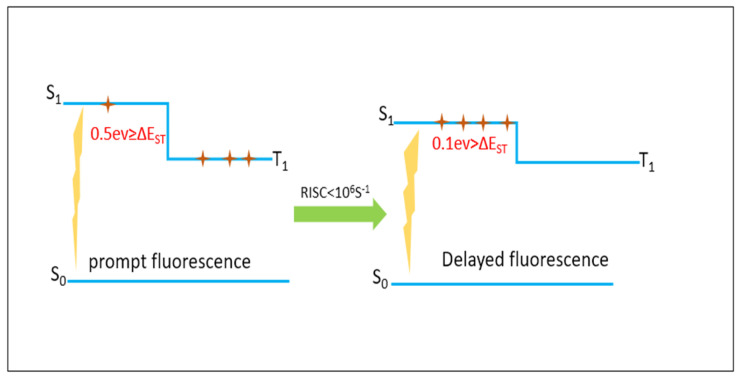
Schematic diagram of TADF mechanism.

**Figure 2 molecules-28-06952-f002:**
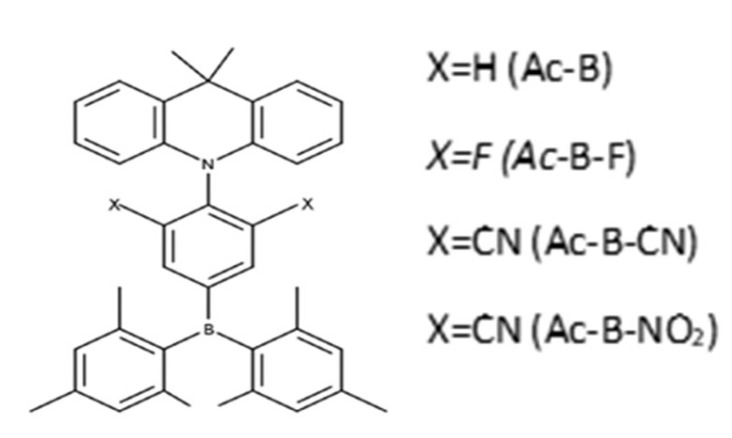
Structures of target molecules.

**Figure 3 molecules-28-06952-f003:**
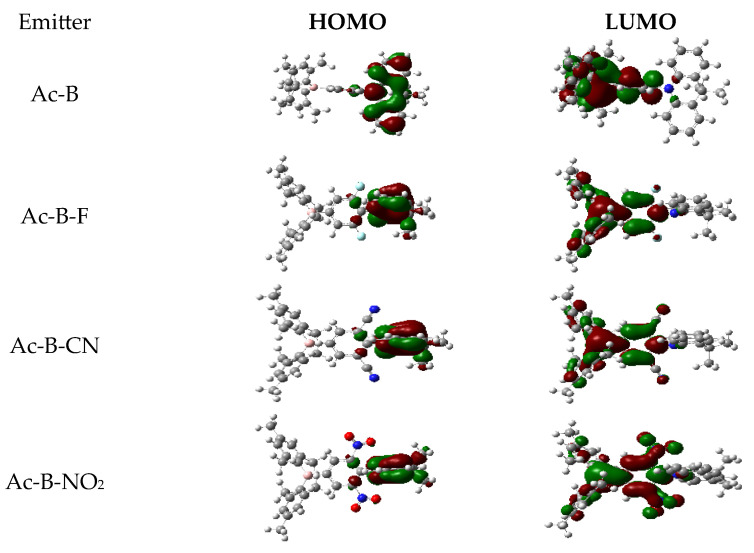
A visualization of HOMOs and LUMOs based on optimized ground state (S_0_).

**Figure 4 molecules-28-06952-f004:**
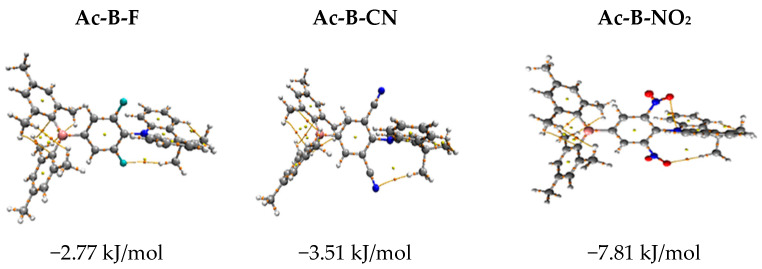
Hydrogen bond formation in Ac-B-F, Ac-B-CN, and Ac-B-NO_2_.

**Figure 5 molecules-28-06952-f005:**
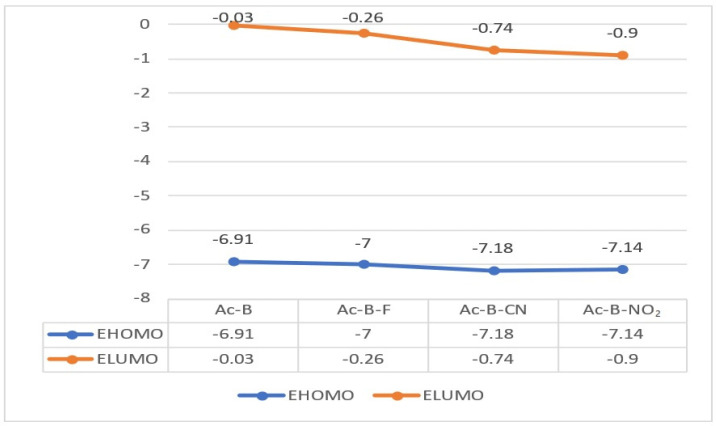
Energies of HOMOs and LUMOs (eV) of all emitters.

**Figure 6 molecules-28-06952-f006:**
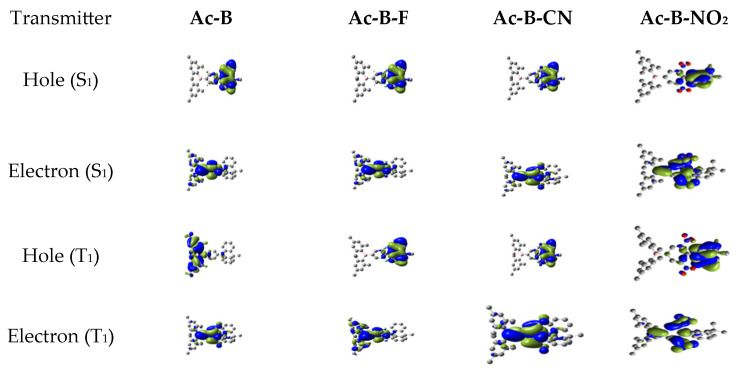
NTOs of S_1_ and T_1_ based on the optimized S_0_ state.

**Figure 7 molecules-28-06952-f007:**
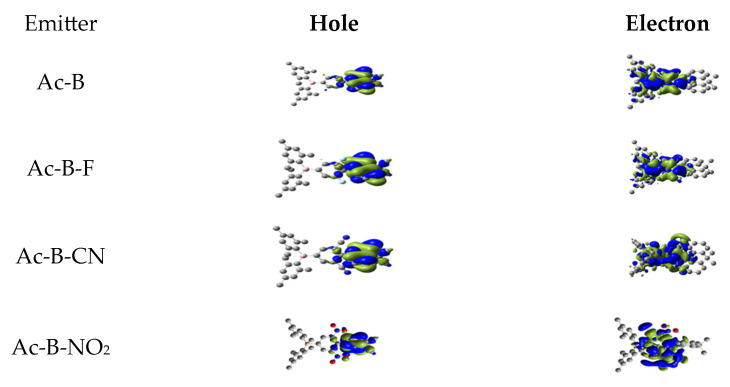
NTOs of the S_1_ state based on its optimized geometry.

**Table 1 molecules-28-06952-t001:** The three lowest singlet and triplet excited states, S_1_, S_2_, and S_3_ and T_1_, T_2_, and T_3_ in eV, respectively, of the target emitters. Vertical (V) energy gaps between S_1_ and T_1_ in eV. The reversed intersystem crossing rate constant (K_RISC_ s^−1^) and radiative decay rate constant (Kr s^−1^) of the studied emitters.

Emitter	S_1_	S_2_	S_3_	T_1_	T_2_	T_3_	∆E_ST_ (V)	∆E_ST_	K_RISC_	Kr
Ac-B	3.48	3.91	4.12	3.43	3.46	3.51	0.05	(0.04) #	3.12 × 10^4^	2.4 × 10^6^
Ac-B-F	3.36	3.83	4.04	3.34	3.36	3.45	0.02	-	2.09 × 10^6^	4.0 × 10^6^
Ac-B-CN	2.71	3.10	3.67	2.69	3.07	3.26	0.02	-	3.50 × 10^6^	1.0 × 10^5^
Ac-B-NO_2_	2.33	2.59	3.41	2.29	2.55	3.24	0.04		1.26 × 10^7^	7.2 × 10^6^

# Taken from [29].

**Table 2 molecules-28-06952-t002:** Oscillator strengths and wavelengths in the S1, S2, and S3 states in all emitters.

Emitter	f (S_1_/S_2_/S_3_)	λ (nm) (S_1_/S_2_/S_3_)
Ac-B	0.0015/0.1255/0.1331	355/317/301 (295/360) **
Ac-B-F	0.0027/0.1370/0.1023	369/324/307
Ac-B-CN	0.0001/0.0004/0.1293	457/400/337
Ac-B-NO_2_	0.0101/0.0018/0.0385	531/478/363

** Taken from [29]

**Table 3 molecules-28-06952-t003:** Adiabatic energy gaps (ad), decay radiative rate constant (Kr), and energy of hydrogen bond for the emitters based on optimized S_1_ state.

Emitters	Adiabatic Energy Gap (S_1_−T_1_) (eV)	Kr (s^−1^)	H-Bond Energy (kJ/mol)
Ac-B	0.02	3.2 × 10^5^	-
Ac-B-F	0.06	5.8 × 10^5^	-
Ac-B-CN	0.01	3.5 × 10^4^	−0.20
Ac-B-NO_2_	−0.04	1.8 × 10^5^	−1.66

**Table 4 molecules-28-06952-t004:** Interfragment charge transfer (IFCT) quantitative analysis of charge transfer (CT) and local excitation (LE) within the lowest excited states, S_1_ and T_1_.

Substrate	S_1_		T_1_	
	LE%	CT%	LE%	CT%
Ac-B	19.21	80.79	77.08	22.92
Ac-B-F	05.07	94.93	07.37	92.63
Ac-B-CN	06.53	93.47	06.70	93.30
Ac-B-NO_2_	07.37	92.63	08.66	91.34

## Data Availability

The data presented in this study are openly available at www.mdpi.com (accessed on 10 September 2023).

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
