# Peer review of "Organoboron Complexes as Thermally Activated Delayed Fluorescence (TADF) Materials for Organic Light-Emitting Diodes (OLEDs): A Computational Study"

_molecules, 2023, doi:10.3390/molecules28196952_

Round 1
Reviewer 1 Report
The manuscript titled "Organic Thermally Activated Delayed Fluorescence (TADF) Materials for Organic Light-Emitting Diodes (OLEDs): A computational Study" reports theoretical calculations of organoboron complexes with small triplet-energy gaps. In general, the manuscript is well written and I would recommend its publication in molecules after the authors address some concerns.
The manuscript focuses on fluorescence emission but the first part of the calculations and results are based on the ground state optimised geometry of the four compounds. Seeing the oscillator strengths obtained, one would expect that the molecules would absorbe light and go to the S2 or S3. How do the authors expect the TADF mechanism to occur from there? Would it be an internal conversion to the S1 and then the intersystem crossing to the triplet? If so, it would be necessary to show conical intersections or large non-adiabatic couplings and spin-orbit couplings among the involved states.
More detailed comments are listed below:
1. The title is too general and the inclusion in it of organoboron complexes would be advised.
2. The supporting information / datasets of the calculations are not available to download
3. Some of the affiliations list "Chemsity" instead of chemistry
4. The abstract should include TADF since it is the main goal of the manuscript
5. Have these molecules been synthetised experimentally? Or why did the authors calculated these four and not others?
6. Figure 3 and table 1 should be connected, i.e., the electronic character of the states S1-S3 needs to be listed (and probably more orbitals should be depicted)
7. For the S1 optimised geometries, a picture of the molecules including dihedral angles would give information on how the charge transfer character changes on that state.
8. In the conclusions part, the authors should mention that excited state dynamics would be needed to unravel the deactivation mechanism of the molecules on their excited states instead of using static approaches.
Author Response
Dear Reviewer
Thank you for your comments. I have uploaded my responses to your comments point by point.
Reviewer 2 Report
Osman et al., reported organoboron complexes with small ΔEST ranging between 0.01 to 0.06 eV. Then the calculated radiative decay rates of ca. 106 s -1. Even though authors calculated some parameters, I recommend the following comments to be addressed. Reviewer feels that major revision is required.
1. In figure 3, the electron density for LUMO is not uniform. Mainly for Ac-B.
2. The ΔEST for Ac-B-CN is very low at 0.02 eV, but radiative decay rate is very less compare to the Ac-B-F. Why? Is there any effects present?
3. Authors should cite some of the state of the art TADF papers : Nat Commun 7, 13680 (2016), Adv. Funct. Mater. 2022, 32, 2110356 ; Adv. Mater., 25: 3707-3714 ; Adv. Sci., 3: 1600080.
4. The calculated Absorption UV-Vis spectra is missing.
5. The theoretical method should be provided first before discussion.
6. The authors mentioned that they are TADF candidates, Howver there is no calculations related to RISC rates, which is key for TADF.
7. Authors should calculate the TRPL and PLQY, non-radiative decay rates, and excited states desnsities to mention the LE/CT states.
Author Response
Dear Reviewer
Thank you for your invaluable comments. They added value to our manuscript..

Round 2
Reviewer 1 Report
The authors have addressed almost all the concerns. However, I still think that the title should include organoboron complexes not to confuse the readers. After that, I would recommend the article suitable for publication.
Reviewer 2 Report
it can be accepted in its present form.
